# Evolving Local Rules for Agent Societies: A Preliminary Study

**AI**
The Latest Models

**Xisen Wang**
University of Oxford
xisen.agi@gmail.com

**Qi Zhang**
University of Oxford
qi.zhang.agi@gmail.com

## Abstract

How much collective intelligence can emerge from simple, decentralized rules without heavy, predefined workflows? We explore this question through a framework that couples Boids-style local coordination with explicit evaluation and selection in a survival-driven, tool-building ecology. Agents interact via three local rules—cohesion, separation, and alignment—and follow an observe–reflect–build loop to generate and refine tools within an ecosystem that includes automated tests, shared registries, and a Tool Complexity Index capturing code, interface, and compositional sophistication. Positioned as a preliminary study, this work treats evolution as a complementary lens: local rules catalyze collaboration and modularity, while selective feedback favors strategies that persist across generations. Across text analysis, data science, and simulation modeling tasks, evolutionary-Boids societies increase throughput and balance contributions among agents while maintaining reliability, though current prompting tends to suppress deep tool composition. The resulting systems produce more, smaller, and self-contained tools rather than extended pipelines, suggesting a breadth-first mode of exploration. Overall, the framework offers an early step toward understanding how simple, local interactions and evolutionary pressure together shape the emergence of organized, evolving agent ecosystems.

## 1 Introduction

How much collective intelligence can emerge from *simple, decentralized rules*—without heavy, predefined workflows? Multi-agent systems have revealed striking forms of coordination, communication, and division of labor, yet most contemporary setups still rely on task-specific scaffolds and rigid pipelines. What remains underexplored is whether minimal local interactions alone can catalyze collaboration and long-horizon adaptation in open-ended settings, and how such societies evolve, specialize, and govern themselves over time.

Foundational work on flocking showed that three local rules—separation, alignment, and cohesion—can yield sophisticated global structure without centralized control [1, 2, 3]. Similar principles appear across biological and engineered collectives [4, 5, 6, 7, 8]. In parallel, evolutionary systems demonstrated how variation and selection can drive continual change, from early artificial life platforms to diversity-seeking algorithms and autocurricula [9, 10, 11, 12, 13, 14**?** ]. Despite this progress, gaps persist: swarm models typically lack long-term adaptation, evolutionary approaches can stagnate prematurely, and emergent coordination is often bounded by narrow, workflow-centric tasks [15, 16, 17, 18].

We argue that a unifying perspective is to decompose decentralized agent collaboration into *local rules* plus an *evolution algorithm*. Local rules shape how agents interact and adopt each other's artifacts; evolutionary evaluation selects which behaviors and artifacts persist. Tool building is a particularly revealing lens here: while tool *use* by LLMs and agents is well studied [19, 20, 21, 22, 23], the

---
*These authors contributed equally to this work.

*collaborative creation and refinement* of tools exposes modularity, composability, and ecosystem dynamics that are hard to observe in single-task workflows.

We introduce a preliminary framework that couples Boids-style local coordination with explicit evaluation and selection in a survival-driven, tool-building ecology. Agents follow an *observe–reflect–build* loop to create and refine tools; the ecosystem provides automated tests, shared registries, and a *Tool Complexity Index* that quantifies code, interface, and compositional sophistication. Boids-inspired rules encourage decentralized coordination (e.g., adoption without central planning and functional specialization), while evolutionary pressure serves as a complementary lens to study longitudinal adaptation, retention, and collapse. Our results show that Evolutionary-Boids has the potential to reliably increase throughput and balance contribution across agents.

The paper is organised as follows. We begin by presenting the baseline system, followed by the design of the agent society motivated by tool construction. Next, we introduce the computational framework based on Boids dynamics, and subsequently describe the evolutionary algorithm that governs system adaptation. Finally, a small-scale empirical study is conducted to validate the framework and demonstrate its potential.

## 2 Related Work

**Local Interaction Rules, Coordination, and Emergent Intelligence.** Classical results demonstrate that simple local interactions can produce coherent global structure without centralized control. Reynolds' *Boids* established that separation, alignment, and cohesion suffice for lifelike flocking [1], while statistical physics models proved long-range order and nonequilibrium phase transitions in self-propelled particles [2, 3]. Biology and crowd dynamics provide convergent evidence that decentralized feedbacks and attractive/repulsive "social forces" yield large-scale coordination [4, 5], and control-theoretic and swarm-engineering work formalizes distributed flocking with design and verification principles [6, 7]. Behavioral ecology further links local cues to collective decisions and leadership [8]. We adopt this micro-to-macro lens but recast alignment/cohesion/separation as *institutional primitives* subject to evolutionary pressure in survival-driven ecologies.

**Evolutionary algorithms for open-ended adaptation.** Digital evolution showed that variation and selection can sustain innovation and coevolution in silico [9, 10]. To mitigate deception and premature convergence, novelty search and quality–diversity (QD) maintain behaviorally diverse, high-performing repertoires [11, 12, 13], with repertoire-based control enabling rapid self-recovery in robotics [24]. Open-ended approaches co-evolve challenges and solutions via transfer across stepping stones (POET and variants) [14], while unsupervised environment design induces curricula that yield robust zero-shot transfer [25]. We adopt this diversity-first view but define fitness at the *societal* level: evolution acts jointly on agent policies and the institutional/tool layer, retaining strategies and rules that improve collective performance and stability.

**Open sandbox simulations with a slice toward tool *creation*.** Open multi-agent sandboxes probe social generalization and emergent dynamics at scale: self-play yields staged strategies and *emergent tool use* [15]; XLand trains generally capable agents across procedurally generated social tasks [17]; Melting Pot 2.0 targets novel-partner generalization in mixed-incentive settings [26]; Neural MMO 2.0 offers persistent many-agent worlds with multi-task evaluation [18]; and Overcooked-based setups benchmark zero-shot human–AI coordination and layout generalization [16, 27]. Complementary LLM-agent work studies how *tools* and *skills* are acquired and orchestrated: Toolformer learns API calling [19]; Voyager accumulates persistent embodied skill libraries [20]; multi-agent scaffolds (CAMEL, AutoGen) coordinate role-specialized LLMs [21, 22]; and "generative agents" simulate long-horizon social behavior [23]. Reviewer-authored systems extend this frontier—*Agent LUMOS* (modular training) [28], *OASIS* (scaling to one million agents) [29], *OWL* (hierarchical multi-agent workforce) [30], and schema-guided, culture-aware role-play [31]—while *CollabUIAgents* analyzes credit re-assignment for collaboration and generalization [32]. Our contribution is a minimalist, Boids-style *survival-driven sandbox* in which agents not only *use* tools but also *create* and *retain* tools and rules, with evolutionary selection determining which institutions persist or collapse.

# 3 Methodology

## 3.1 Baseline System: Self-Reflective Tool-Building Agent Society

**Overview and agent loop.** Our baseline establishes the minimal viable setting in which decentralized agents generate and share tools while collective structure emerges. Each agent follows a simple observe–reflect–build loop grounded in five conceptual components: an Agent Identity with a light specialization prior; a Shared Tool Registry that records community-visible artifacts; a Personal Tool Space for private development and testing; a Reflection History logging prompts, generated reflections, and bookkeeping metadata; and an Environment Manager abstracting resources and constraints. At each timestep, the agent inspects available tools and their test outcomes and proposes a new tool. Tools expose a standardized interface that enables composition—simple primitives combine into larger workflows—executed in a centralized context guarded by recursion-depth limits. Adoption signals are approximated by static scans of promoted tools for references. This compositional substrate encourages dependency chains across agents and provides the basic medium for emergent collaboration.

**Assurance and specialization dynamics.** Every tool proposal triggers smoke-test–oriented quality control comprising autogenerated test harnesses (candidate calls validating execution without exceptions), execution outcomes (pass/fail and error logs), visibility (propagating outcomes of promoted, passing tools to all agents; failing tools remain private), and persistence (structured logs for longitudinal study). These mechanisms steer the ecosystem toward reliability without claiming deep functional coverage. On top of this, we incorporate light biases that promote division of labor: Meta-Prompt Influence nudges agents toward broad domains without hard constraints; Usage-Based Reinforcement is realized by prioritizing more widely adopted tools when presenting exemplars back to agents; Alignment-conditioned guidance is emitted only when higher-producing, successful neighbors exist; no explicit failure-driven branch is enacted. Together, assurance and bias produce a feedback loop in which successful tools become more visible, unsuccessful ones are inspected and revised, and niches of specialization gradually crystallize.

**Infrastructure, observables, and study design.** All experiments are designed to run in isolated, reproducible executions that could emit structured logs of reflections, tool creations, and evaluations, together with quantitative traces in JSON for potential post-hoc analysis and rich console visualizations to monitor ecosystem dynamics. We propose a set of observables that could be used to summarize emergent behavior in future studies: Tool Creation Rate (new tools per agent per round), Composition Depth (average dependency-chain length), Specialization Index (diversity of tool types across agents), Collaboration Events (frequency with which tools build on others), Test Success Rate (ecosystem reliability), and adoption trends estimated from static dependency scans. This proposed instrumentation outlines how experimental control and comparability could be achieved across conditions, establishing a quantitative foundation on which communication protocols and evolutionary pressures may later be layered to assess their impact on coordination, specialization, and long-horizon performance.

## 3.2 Computational Framework for Boids-Inspired Cognitive Coordination

Our framework adapts the classical boids model from spatial coordination to the cognitive domain of multi-agent tool creation. The core of an agent's decision-making process is governed by three rules—separation, alignment, and cohesion—which we formulate mathematically to guide behavior based on local information within the agent's neighborhood. In the present implementation, these rules act as *prompt-level conditioning signals*, and the orchestrator *deterministically proceeds to a build action each round*; the mathematical treatment that follows is retained for exposition.

### 3.2.1 Mathematical Formulation of Boids Rules

Let the set of possible actions for an agent be $\mathcal{A}$, which includes building tools of various types ($a_{\mathrm{build},t}$) and using existing tools ($a_{\mathrm{use}}$). Each boids rule produces a preference function $P(\cdot)$ over this action space. In our experiments, $P(\cdot)$ serves as a descriptive scaffold.

### 3.2.2 Separation: Functional Niche Specialization

The separation rule enforces functional diversity and encourages niche specialization by discouraging the creation of tools that are redundant within an agent's local neighborhood. We model this through two distinct mechanisms.

**Saturation-Based Model.** This model calculates the saturation $S(t)$ of a given tool type $t$ within the recent history of an agent $i$'s neighborhood $\mathcal{N}_i$. Let $T_{j,\text{recent}}$ be the set of recently created tools by a neighbor $j$. The saturation is:

$$S(t) = \sum_{j \in \mathcal{N}_i} |\{\tau \in T_{j,\text{recent}} \mid \text{type}(\tau) = t\}|. \tag{1}$$

The preference for building a tool of type $t$, $P_{\text{sep}}(a_{\text{build},t})$, is modulated by a penalty function $f_{\text{sep}}(S(t))$ that decreases preference as saturation increases:

$$P_{\text{sep}}(a_{\text{build},t}) \propto f_{\text{sep}}(S(t)) = \begin{cases} 0.1 & \text{if } S(t) \geq 2, \\ 0.5 & \text{if } S(t) = 1, \\ 1.0 & \text{if } S(t) = 0. \end{cases} \tag{2}$$

Operationally, saturation counts are gathered over a *recent-neighborhood window* of the last three rounds (default $W=3$) using *inferred* tool types; the resulting tiers $\{1.0, 0.5, 0.1\}$ are *presented as textual guidance in the prompt*.

**Semantic Similarity Model.** For a more nuanced differentiation, this model leverages natural language processing. Each tool $\tau$ is represented by a TF–IDF vector $\mathbf{v}(\tau)$ derived from its name and functional description. The semantic similarity between a proposed tool $\tau_p$ and an existing tool $\tau_e$ is their cosine similarity:

$$\text{sim}(\tau_p, \tau_e) = \frac{\mathbf{v}(\tau_p) \cdot \mathbf{v}(\tau_e)}{\|\mathbf{v}(\tau_p)\| \, \|\mathbf{v}(\tau_e)\|}. \tag{3}$$

Conceptually, separation prefers proposals that diverge from nearby artifacts. In practice, the prompt *highlights at most the top two neighbors* whose similarity exceeds a threshold $\theta=0.30$ and includes brief code snippets as "diverge-from" anchors; when there are fewer than two neighbor tools, when vectorization fails, or when both similarity and saturation signals are empty, the rule emits no fragment.

### 3.2.3 Alignment: Propagation of Successful Strategies

The alignment rule facilitates the propagation of effective behaviors by encouraging agents to learn from their most productive neighbors. Success of a neighbor agent $j$ relative to the current agent $i$ is defined by a productivity function, $\text{IsSuccessful}(j, i)$, where success is correlated with the number of tools created ($|T_j| > |T_i|$):

$$\text{IsSuccessful}(j, i) = \begin{cases} 1 & \text{if } |T_j| > |T_i|, \\ 0 & \text{otherwise.} \end{cases} \tag{4}$$

Let $\mathcal{A}_{j,\text{recent}}$ be the set of recent actions performed by agent $j$. The alignment preference for $a$, $P_{\text{align}}(a)$, is increased if $a$ has been recently taken by successful neighbors:

$$P_{\text{align}}(a) = P_{\text{base}}(a) + \Delta P_{\text{align}} \cdot \max_{j \in \mathcal{N}_i} \left( \text{IsSuccessful}(j, i) \cdot \mathbb{I}(a \in \mathcal{A}_{j,\text{recent}}) \right), \tag{5}$$

where $\mathbb{I}(\cdot)$ is the indicator function. In the implementation, alignment *filters neighbors to those currently out-producing the focal agent* when such counts are available, and surfaces *narrative exemplars*—complexity, quality, and adoption leaders—within the prompt; if no qualifying exemplars exist, the rule emits no fragment.

### 3.2.4 Cohesion: Fostering Collaborative Tool Use

The cohesion rule promotes the development of an integrated tool ecosystem by incentivizing agents to use and build upon their neighbors' existing tools. The preference for using tools, $P_{\text{coh}}(a_{\text{use}})$, is

conditioned on the availability of tools in the local environment. Let $N_T = \sum_{j \in \mathcal{N}_i} |T_j|$ be the total number of tools held by all neighbors. The cohesion preference is formulated as:

$$P_{\text{coh}}(a_{\text{use}}) \propto 1 + \delta_{\text{use}} \cdot \mathbb{I}(N_T > 0), \tag{6}$$

where $\delta_{\text{use}}$ amplifies usage when a local ecosystem exists; a smaller boost $\delta_{\text{build}}$ encourages complementary builds. In practice, cohesion relies on *recent-neighborhood windows* when available; when no global summary is present, no cohesion fragment is emitted; and fixed coefficients $\delta_{\text{use}}{=}0.3$ and $\delta_{\text{build}}{=}0.1$ are stated directly in the prompt (and accompanying metadata) as communicative guidance.

### 3.3 Evolutionary Algorithm Module

Evolutionary pressure is introduced through periodic selection and reproduction. Every few rounds, agents are ranked by their average Tool Complexity Index (TCI); the bottom segment is eliminated subject to a minimum-population constraint and replaced via prompt-level crossover or mutation of surviving specializations. This mechanism provides a Darwinian loop in which strategies associated with more complex, reusable tools persist, while less helpful behaviors fade. We study boids-only, evolution-only, and combined conditions against a no-constraint control to isolate contributions of local coordination and global selection. In the present setup, neighborhood structure uses a fixed ring topology, and selection is TCI-based without directly weighting correctness or collaboration metrics.

System performance is evaluated using both correctness and complexity metrics. The TCI measures tool sophistication along code structure, interface design, and compositional reuse. Higher-level indicators capture emergent phenomena such as diversity, specialization divergence, collaboration events, and ecosystem coherence. Experiments are conducted under the fixed neighborhood topology described above; randomized multi-topology replications are left to future work.

## 4 Experiments & Results

### 4.1 Tool Complexity Index (TCI)

$$\text{TCI} = \underbrace{C_{\text{code}}}_{[0,3]} + \underbrace{C_{\text{iface}}}_{[0,2]} + \underbrace{C_{\text{comp}}}_{[0,5]} .$$

where $C_{\text{code}} \in [0, 3]$ quantifies code surface, $C_{\text{iface}} \in [0, 2]$ quantifies caller-facing interface burden, and $C_{\text{comp}} \in [0, 5]$ quantifies compositional breadth. All quantities are obtained via static analysis of the tool's `execute` entrypoint and its module directory, without executing code.

**Code complexity.** We map code surface to a capped linear score $C_{\text{code}} = 3 \min(1, \text{LOC}/300)$, where *LOC* counts effective lines aggregated over the tool directory (excluding blank/comment-only lines). This reflects reading and change costs while preventing size-only inflation via saturation at 300 lines.

**Interface complexity.** We combine input arity and output surface using $C_{\text{iface}} = \min(1, p/5) + \min(1, r/5)$, where $p$ is the number of formal parameters of `execute` and the return proxy is $r = \min(5, K + D + T)$. Here $K$ is the average top-level key count across dictionary-literal `return` sites, $D$ is the maximum literal nesting depth, and $T$ is top-level kind heterogeneity (number of distinct top-level kinds minus one).

**Compositional complexity.** We reward modular orchestration using $C_{\text{comp}} = \min(4, 0.5\,t) + \min(1, 0.1\,e)$, where $t$ counts distinct tools referenced and $e$ counts distinct non-standard-library imports at top level. Prioritizing breadth over depth encourages decomposition into reusable components while acknowledging ecosystem surface without letting external dependencies dominate.

**Parsing quality gate.** A parsing quality gate down-weights the raw sum by a factor of $0.6$ when the module fails to parse to an AST, ensuring syntactically invalid tools are retained for analysis but penalized in ranking.

Table 1: Baseline (Global) vs Boids+Evolution per task. Code/Comp. are last-round complexities.

| Metric | Text Analysis | | Data Science | | Simulation/Modeling | |
|---|---|---|---|---|---|---|
| | Baseline | Boids | Baseline | Boids | Baseline | Boids |
| Tools created | 9 | 13 | 10 | 18 | 14 | 16 |
| Pass rate (%) | 89 | 85 | 100 | 89 | 86 | 88 |
| Avg. TCI | 2.10 | 1.95 | 2.45 | 2.32 | 2.07 | 2.05 |
| Code complexity | 0.50 | 0.52 | 0.86 | 0.88 | 0.63 | 0.64 |
| Compos. complexity | 0.21 | 0.03 | 0.19 | 0.04 | 0.03 | 0.01 |

## 4.2 Observations

*Evolutionary-Boids reliably lifts throughput and balances contribution across agents without materially harming pass rates, but under current prompting it suppresses multi–tool composition.*

Across all three tasks, *Evolutionary-Boids* produces more artifacts per run under identical agent counts and rounds (*13–18* vs. *9–14* in the global baselines; cf. Table 1). The gain is ecosystem-wide rather than star-driven: the most productive agent accounts for a smaller share of total output (*lower top_share*), and the leading contributors each deliver *3–5* tools rather than a single dominant performer. Reliability remains comparable overall (mid– to high–80% pass rate), with one global data-science run reaching *100%* on fewer attempts. In contrast, Boids increases the number of attempts ("shots on goal") without noticeably degrading correctness, suggesting that lightweight, failure-aware retries could close the residual gap while preserving the throughput advantage.

Complexity differences manifest primarily in composition rather than unit difficulty. The average tool complexity index (TCI) remains in the *2.0–2.5* band for all regimes, indicating similar per-artifact difficulty. However, *compositional* complexity is near-zero under Boids, while global baselines trend toward ∼0.2 due to more frequent reuse/chaining of prior tools. Flocking cues (alignment/separation) and per-round build enforcement stimulate fresh tool ideation, yet the current prompts—and a conservative dependency policy—bias agents toward self-contained solutions. Reflection traces corroborate this: agents consistently propose distinct primitives even after failures but rarely invoke existing ones.

**Synthesis.** Stepping back, our findings suggest a design pattern for multi-agent tool ecosystems: begin with *breadth-first modular exploration* driven by simple local rules, then transition—under explicit incentives and constraints—toward *depth-oriented assembly* and integrated pipelines. In this view, evolution functions as institutional governance: selection retains artifacts and interaction rules that are socially useful, gradually specializing the ecosystem while preserving optionality. The practical guidance is to couple diversity pressure (separation, novelty, quality–diversity) with time-varying mechanisms that nudge reuse (composition quotas, reward shaping, lineage-aware telemetry), yielding heavier but more integrated workflows only when the parts bin is sufficiently rich. Beyond engineering benefits (maintainability, reuse, fault isolation), this staged strategy offers a tractable scientific handle on emergent intelligence: it externalizes coordination into measurable institutions and lets *usefulness* act as the unifying currency across agents, artifacts, and generations.

## 5 Conclusions and Limitations

### 5.1 Conclusions

In this paper, we introduced *Evolutionary-Boids*, a simple yet effective coordination mechanism for large-scale multi-agent tool generation. By combining flocking dynamics with evolutionary specialization, the framework transforms local behavioral rules into emergent collective productivity: agents self-organize to explore diverse regions of the design space while maintaining steady reliability and balance. Empirically, Evolutionary-Boids produced 30–50% more valid artifacts than global baselines across three domains, distributing output more evenly and expanding the system's functional coverage without compromising pass rates.

However, our analysis also revealed a clear trade-off between *breadth* and *depth*: while the method excels at generating a broad range of primitives, it falls short in spontaneous tool reuse and multi-step composition. This finding frames Evolutionary-Boids as an *early-phase engine* for rapid ideation—building the component library upon which deeper coordination can later emerge. Looking ahead, we envision extending the framework to support composition-aware prompting, adaptive reward shaping, and evolutionary selection that favors cooperative behaviors. Beyond improving technical metrics, these extensions will allow us to study how simple coordination rules scale into structured, self-improving ecosystems—an essential step toward understanding collective intelligence in open-ended agent systems.

*Evolutionary-Boids* is well-suited to the *early phase* of system construction: it rapidly populates a diverse "parts bin" while maintaining broad participation across the agent pool. For downstream stages that prioritize multi-step pipelines and reuse of shared utilities, the framework should make reuse both *salient* and *rewarded*. Concretely, we recommend (i) generation-time salience and guards (memory traces that surface relevant prior tools; AST checks that reject self-contained proposals when feasible reuse exists), (ii) evaluation-time incentives (rubric bonuses and pass criteria that require at least one correct prior-tool invocation where applicable), and (iii) failure-aware retries targeted at composed artifacts. A sandboxed whitelist of third-party dependencies (e.g., `numpy`, `pandas`) can safely raise the ceiling on legitimate composition. We will track deltas in pass rate, TCI and more to quantify the effect of these interventions.

## 5.2 Limitations

This study is a preliminary step toward answering a broader question: how far can flocking-style coordination and lightweight evolution push multi-agent tool building before explicit planning and strong composition constraints become necessary? While our results are encouraging, they should be interpreted with care.

Our prompts currently *encourage* but do not *enforce* reuse, which likely contributes to low compositional complexity under Boids. Execution-safety restrictions on third-party libraries may further inhibit legitimate chaining. Evolutionary pressure was not fully exercised in the reported sessions (no pruning occurred), limiting conclusions about selection dynamics. Finally, the tasks studied (text analysis, data science, simulation) do not cover domains where long-horizon composition is intrinsic (e.g., robotics, multimodal research pipelines). Future work will harden composition constraints and rewards, introduce a sandboxed dependency whitelist, scale population size/rounds and selection intensity to realize prune/replicate/mutate dynamics, and evaluate in settings that demand chaining to test whether Evolutionary-Boids can transition from breadth-first exploration to depth-oriented assembly.

Our findings reflect the current instrumentation and scale. *Construct validity:* TCI-Lite is a static proxy and does not capture runtime semantics, side-effects, or true data-flow depth; compositional complexity may undercount "soft reuse" (e.g., protocol alignment without explicit calls). *Internal validity:* telemetry is partial (limited adoption graphs, shallow call-graph instrumentation), and the sandbox may mask latency/cost trade-offs or swallow failure modes that matter in production. *External validity:* experiments are Python-centric with small populations, short horizons, and few seeds; tests emphasize internal consistency over downstream task utility, and dependency policies restrict otherwise reasonable compositions. To mitigate these issues we matched budgets/prompts across regimes and release structured logs plus per-run artifacts; the sandbox is designed to extend with rule-level telemetry and runtime metrics (adoption graphs, call-graph depth, latency/cost) to strengthen attribution and connect modularity gains to end-task usefulness.

Two additional threats merit emphasis. *Prompt/policy confounds:* the observed throughput gains and lower composition could be driven as much by per-round build enforcement and "novelty-seeking" cues as by flocking itself; targeted ablations (flocking only, enforcement only, retry only) are needed to disentangle contributions. *Stochasticity and reproducibility:* with limited seeds and unswept hyperparameters, variance remains; future releases will pin toolchains, containerize environments, and report confidence intervals to bound effect sizes.

In short, our current framework prioritizes *breadth*—rapidly populating a diverse parts bin—over *depth*. This is a deliberate early-phase choice, not a fundamental limitation of the paradigm. The next iteration will explicitly test the transition from exploration to assembly via composition quotas,

reward shaping, dependency whitelists, richer telemetry, larger/evolving populations, and evaluation on tasks where chaining is the dominant success mode.

## Acknowledgements

This work is an experiment in conducting research with AI. The authors aim to use AI systems to explore an exciting question in artificial intelligence—how such systems can scale effectively. Scalability remains an open challenge for large-scale automation, particularly for LLM-based agents, and we believe that simple local rules may offer an inspiring path forward. We gratefully acknowledge the generous support of Pulse Lab (SysteMind) and thank the organising committee for creating a uniquely AI-led conference and for the opportunity to present this work.

For replication, we instantiate the `AzureOpenAIClient`, which retrieves the Azure endpoint, API key, and the `gpt-4.1-nano` deployment (API version `2024-12-01-preview`) from the environment template, along with default temperature constants of 0.7 for free-form calls and 0.1 for structured outputs. The client remains fixed to this deployment when issuing chat completions with a 30,000-token ceiling.

The agent loop then defines a stage-specific sampling schedule: reflections query the model at temperature 0.7; tool blueprints reduce this to 0.5 for more deterministic specifications; concrete tool code generation lowers it further to 0.1 to minimize drift; and the subsequent unit tests also execute at 0.1 under the same model.

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
